# High-Performance ε-Ga₂O₃ Solar-Blind Photodetectors Grown by MOCVD with Post-Thermal Annealing

Zeyuan Fei [1], Zimin Chen [1,*], Weiqu Chen [1], Tiecheng Luo [1], Shujian Chen [1], Jun Liang [1,2,*], Xinzhong Wang [2], Xing Lu [1], Gang Wang [1,3] and Yanli Pei [1,3,*]

[1]  State Key Laboratory of Optoelectronic Materials and Technologies, School of Electronics and Information Technology, Sun Yat-sen University, HEMC, Guangzhou 510275, China; feizy@mail3.sysu.edu.cn (Z.F.); chenwq59@mail2.sysu.edu.cn (W.C.); luotch3@mail2.sysu.edu.cn (T.L.); chenshj65@mail2.sysu.edu.cn (S.C.); lux86@mail.sysu.edu.cn (X.L.); stswangg@mail.sysu.edu.cn (G.W.)

[2]  Shenzhen Institute of Information Technology, Shenzhen 518172, China; wangxz@sziit.com.cn

[3]  Foshan Institute, Sun Yat-sen University, Foshan 528225, China

*   Correspondence: chenzim8@mail.sysu.edu.cn (Z.C.); liangjun@sziit.edu.cn (J.L.); peiyanli@mail.sysu.edu.cn (Y.P.)

**Abstract:** High-temperature annealing has been regarded as an effective technology to improve the performance of Ga₂O₃-based solar-blind photodetectors (SBPDs). However, as a metastable phase, ε-Ga₂O₃ thin film may undergo phase transformation during post-annealing. Therefore, it is necessary to investigate the effect of the phase transition and the defect formation or desorption on the performance of photodetectors during post-annealing. In this work, the ε-Ga₂O₃ thin films were grown on c-plane sapphire with a two-step method, carried out in a metal-organic chemical vapor deposition (MOCVD) system, and the ε-Ga₂O₃ metal-semiconductor-metal (MSM)-type SBPDs were fabricated. The effects of post-annealing on ε-Ga₂O₃ MSM SBPDs were investigated. As a metastable phase, ε-Ga₂O₃ thin film undergoes phase transition when the annealing temperature is higher than 700 °C. As result, the decreased crystal quality makes an SBPD with high dark current and long response time. In contrast, low-temperature annealing at 640 °C, which is the same as the growth temperature, reduces the oxygen-related defects, as confirmed by X-ray photoelectron spectroscopy (XPS) measurement, while the good crystal quality is maintained. The performance of the SBPD with the post-annealing temperature of 640 °C is overall improved greatly compared with the ones fabricated on the other films. It shows the low dark current of 0.069 pA at 10 V, a rejection ratio ($R_{peak}/R_{400}$) of $2.4 \times 10^4$ ($R_{peak} = 230$ nm), a higher photo-to-dark current ratio (PDCR) of $3 \times 10^5$, and a better time-dependent photoresponse. These results indicate that, while maintaining no phase transition, post-annealing is an effective method to eliminate point defects such as oxygen vacancies in ε-Ga₂O₃ thin films and improve the performance of SBPDs.

**Keywords:** solar-blind photodetector; ε-Ga₂O₃; annealing; MOCVD

## 1. Introduction

Solar-blind photodetectors (SBPDs) have garnered considerable attention for their applications in missile warning, optical communication, environmental monitoring, flame detection, and various other fields [1–3]. Recently, solar-blind UV photodetectors based on ultrawide bandgap semiconductors like AlGaN, MgZnO, and Ga₂O₃ have gained significant prominence [4–6]. Among these materials, Ga₂O₃ stands out as a promising candidate for SBPDs due to its ultra-wide and direct bandgap (4.5–5.2 eV), which is suitable for the solar-blind region without the need for bandgap modulation through alloying processes [7]. Ga₂O₃ has six crystal phases: α, β, γ, δ, ε, and κ [8,9]. Among them, β-Ga₂O₃, being the most stable phase, has garnered widespread attention for the fabrication of SBPDs [10–12]. Metastable ε-Ga₂O₃ presents a hexagonal symmetry, which means ε-Ga₂O₃ has potential to grow on hexagonal substrates such as SiC, sapphire, and GaN with

high-quality heteroepitaxy [13–15]. It also means that $\varepsilon$-$Ga_2O_3$ grown on these matching substrates is attractive for the development of novel devices for optoelectronic applications.

$Ga_2O_3$ SBPDs can be categorized into two basic types: photoconductive-metal–semiconductor-metal (MSM) SBPDs and junction SBPDs. The MSM-SBPDs consist of two "back-to-back" interdigitated electrodes and a semiconductor with photoconductivity effect as the work mechanism. This type of device offers a simple fabrication process and large responsivity [16]. The impact of defects in semiconductors on the performance of MSM-SBPDs can appear in a variety of ways. These defects can be point defects, line defects, and plane defects. An appropriate amount of defects can increase photocurrent due to photoconductivity gain, but a large number of defects can also cause a significant increase in dark current and prolong response time. To date, there exist numerous studies examining the influence of post-annealing on the characteristics of $\beta$-$Ga_2O_3$ film photodetectors [17–20]. High-temperature annealing is recognized as a potent technique capable of not only diminishing defect density but also enhancing crystal quality. These factors significantly influence the performance of solar-blind photodetectors based on $Ga_2O_3$. The post-annealing method has also been applied to the $\varepsilon$-$Ga_2O_3$ photodetectors to modulate the oxygen vacancy in $\varepsilon$-$Ga_2O_3$ film, as reported in the literature [21]. However, as a metastable phase, $\varepsilon$-$Ga_2O_3$ thin film may undergo phase transformation during post-annealing. Therefore, it is necessary to investigate the effect of the phase transition on the performance of photodetectors. In addition, by keeping the $\varepsilon$-$Ga_2O_3$ phase unchanged, we would like to know whether a post-annealing temperature as low as the growth temperature of $\varepsilon$-$Ga_2O_3$ is effective in modulating defects such as oxygen vacancies.

In this study, MSM SBPDs were produced using $\varepsilon$-$Ga_2O_3$ thin films cultivated through metal–organic chemical vapor deposition (MOCVD). The impact of varying annealing temperatures on both the quality of the $\varepsilon$-$Ga_2O_3$ thin films and the performance of SBPDs was thoroughly examined. When undergoing high-temperature annealing, the phase transition results in the SBPDs' performance deterioration, with high dark current and long response time. Compared with high-temperature annealing, post-annealing at temperatures as low as the growth temperature significantly improves the performance of SBPDs. The detailed mechanism is also discussed.

## 2. Experiments

The $\varepsilon$-$Ga_2O_3$ thin films were grown on c-plane sapphire substrates utilizing an Emcore400 metal–organic chemical vapor deposition (MOCVD) system, employing a two-step growth method. Initially, a nucleation layer was grown at 550 °C using Triethylgallium (TEGa) and deionized water ($H_2O$) as precursors. Subsequently, an epitaxial layer was cultivated at 640 °C using the same precursors. The thicknesses of the nucleation layer and epitaxial layer were about 30 nm and 360 nm, respectively. Afterward, the $\varepsilon$-$Ga_2O_3$ films underwent annealing at temperatures of 640, 700, 800, and 900 °C in a tube furnace under an $N_2$ atmosphere. These samples were labeled as AG, N-640, N-700, N-800, and N-900, respectively, denoting different annealing temperatures. The crystal structure of the $Ga_2O_3$ films was scrutinized using high-resolution X-ray diffraction (HRXRD) via a Bruker D8 Discover instrument (Bruker, Mannheim, Germany), using a ThermoFisher Nexsa (ThermoFisher, Waltham, MA, USA) with a monochromic Al Ka (h$\nu$ = 1486.6 eV) X-ray source. Morphological assessments of the $Ga_2O_3$ thin films were conducted using a scanning electron microscope (SEM) (Hitachi S-4800) (Hitachi, Tokyo, Japan) and atomic force microscopy (AFM) (Veeco, Dimension EDGE AFM, Plainview, NY, USA). X-ray photoelectron spectroscopy (XPS) was employed to evaluate the stoichiometry of the $Ga_2O_3$ films.

MSM SBPDs with Ti/Au (20 nm/80 nm) electrodes were fashioned on both the as-grown and nitrogen-annealed $Ga_2O_3$ films. The electrode fingers measured 3 $\mu$m in width and 95 $\mu$m in length, with a 5 $\mu$m spacing gap. An annealing process at 500 °C for 10 min was executed to ensure full contact between the Ti/Au and $Ga_2O_3$ films. Current–voltage (I-V) characteristics and time-dependent photocurrent (I-t) curves were measured using

a semiconductor parameter analyzer (Agilent B1500A) (Agilent, Santa Clara, CA, USA). A photoelectric system (CEL-PF300, China Education Au-light) (China Education Au-light, Beijing, China) emitting light with various power densities ($P_{light}$) and illumination wavelengths (λ) ranging from 220 nm to 400 nm was employed as a light source. Transient response measurements were obtained using a 254 nm lamp with a light intensity of 550 μW/cm².

### 3. Results and Discussion

Figure 1a presents the results of the X-ray diffraction (XRD) 2θ-scan of $Ga_2O_3$ film growth on c-plane sapphire at different thermal annealing temperatures. The observed diffraction peaks at 38.90° and 59.80° correspond to the (004) and (006) planes of ε-$Ga_2O_3$. Additionally, peaks at 37.34°, 38.38°, and 58.94° are associated with the β-$Ga_2O_3$ reflections, specifically ($\overline{3}$11), ($\overline{4}$02), and ($\overline{6}$03), respectively. It is noteworthy that the highest intensity peak aligns with the sapphire (006) plane [22,23]. It is observed that AG and N-640 are nearly pure ε phase without obvious phase transition. The weak shoulders at 38.38° and 58.94° in logarithmic coordinates can be attributed to the seeding layer, which the XRD 2θ-scan showed in Figure S1. Then, obvious diffraction shoulder peaks at 37.34° and 58.94° can be seen in the film with 700 °C thermal annealing, indicating that β-$Ga_2O_3$ component in the film begins to form, which means that a mixture of β- and ε-$Ga_2O_3$ appears at this annealing temperature. When the annealing temperature increases to 800 °C or above, the ε-$Ga_2O_3$ transforms to β-$Ga_2O_3$ completely. We observed diffraction peaks from different crystal plane families, indicating that β-$Ga_2O_3$ is poly-crystalline structure. The transition temperature is consistent with literature reports [24]. In order to evaluate the effect of low-temperature annealing on the crystal quality carefully, the (004) rocking curves of the pure ε-phase thin films were compared between AG and N-640, as shown in Figure 1b. The full width at half maximum (FWHM) of AG is 0.22°, indicating good crystal quality by heteroepitaxy. The FWHM is almost unchanged after 640 °C post-annealing. This suggests that the low-temperature annealing (i.e., the annealing temperature is almost the same as the growth temperature) does not easily create defects such as dislocation. The transmission spectra of all the films are shown in Figure 1c with the wavelength from 200 to 800 nm. By extrapolating the linear region of (αhν)² versus photon energy (hν), a band gap range around 4.73–4.81 eV was determined.

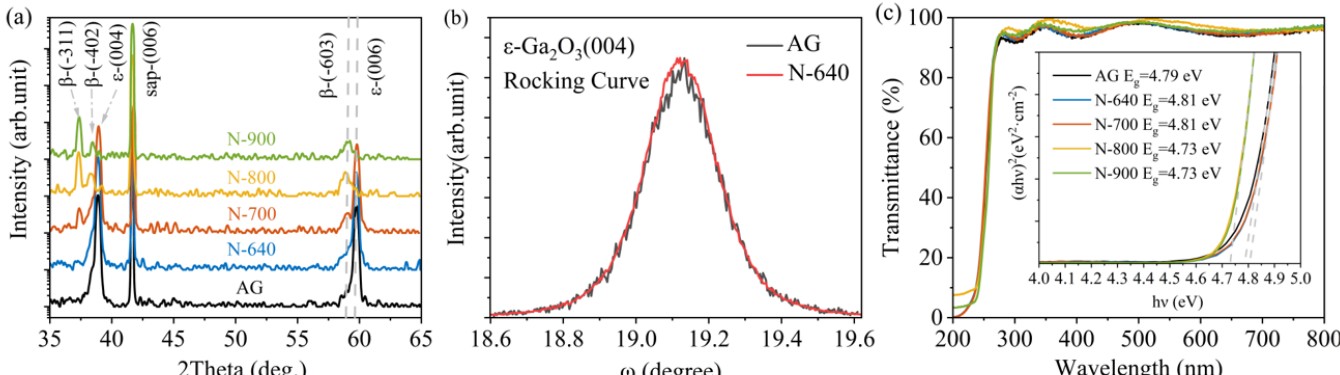

**Figure 1.** (**a**) XRD patterns of $Ga_2O_3$ film growth on c-plane sapphire with various thermal annealing temperatures. (**b**) Rocking curve of the ε-$Ga_2O_3$ (004) plane. (**c**) The transmittance spectra of the $Ga_2O_3$ thin film, a plot of (αhν)² against photon energy (hν), which aids in determining the optical bandgap of $Ga_2O_3$, as presented in the inset.

To further investigate the crystalline structure of the annealed ε-$Ga_2O_3$ films, Figure 2 presents the surface morphology images of the $Ga_2O_3$ thin films obtained through SEM and AFM measurements. The three-dimensional island surface morphologies are observed by AFM for all the films with a Root-Mean-Square (RMS) surface roughness of about 6–7 nm. In addition, as the annealing temperature increased to 800 °C, obvious cracks appeared on

the surface of the film (N-800). Furthermore, with the annealing temperature elevated to 900 °C (N-900), the cracks become wider and wider, which means that the process of phase transition is accompanied with significant stress, leading to deterioration of crystal quality.

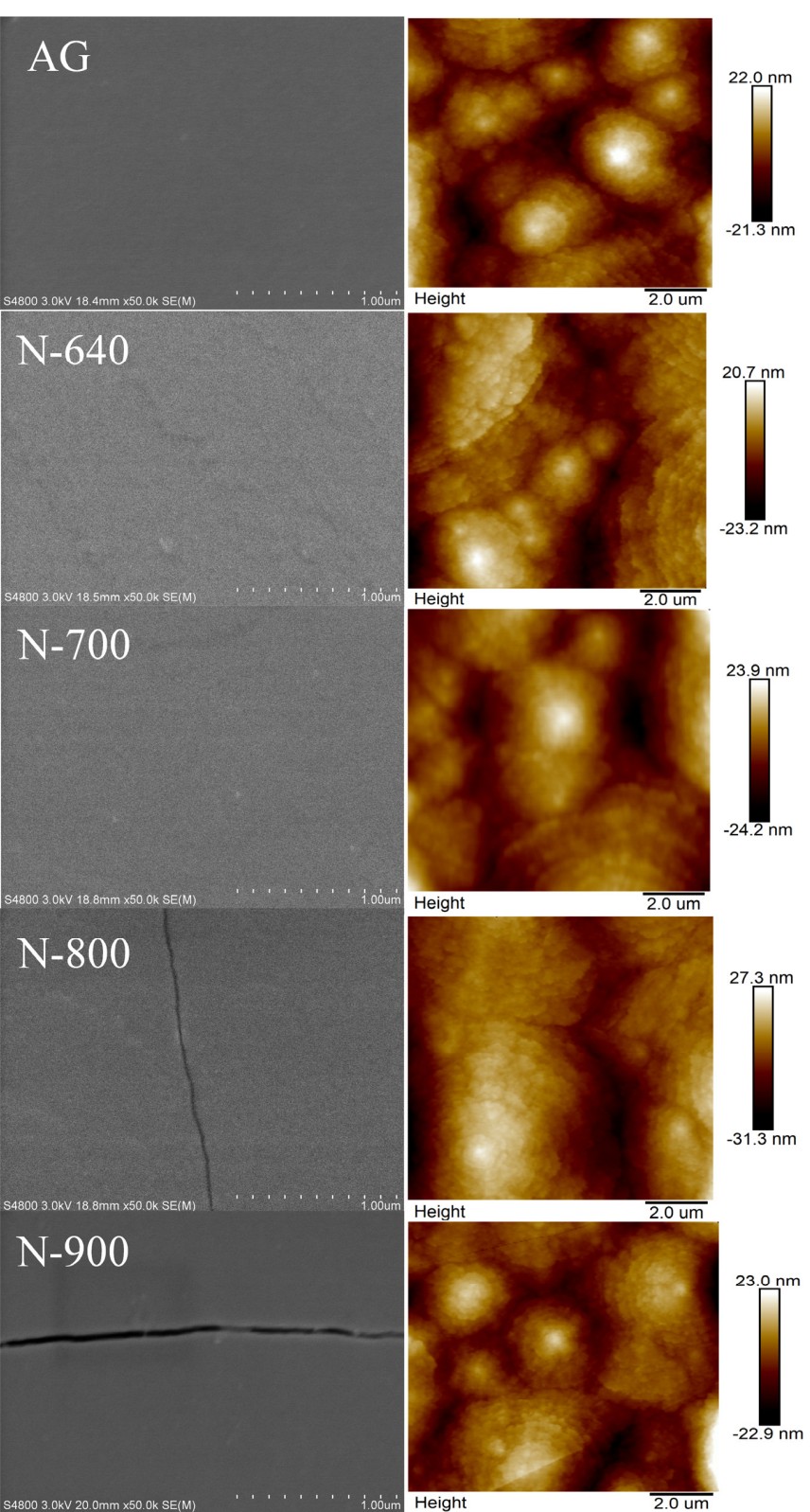

**Figure 2.** SEM top-view (**left** column) and AFM surface morphology images (**right** column) of the Ga$_2$O$_3$ films with various thermal annealing temperatures.

In recent investigations, both empirical and theoretical research has revealed the abundance of oxygen-vacancy (Vo) defects in $Ga_2O_3$ [25,26]. Figure 3a–e show the XPS spectra of the O 1s peaks for both as-grown and annealed ε-$Ga_2O_3$ thin films. The O 1s core level spectra were subjected to Gaussian fitting, revealing three distinct components: (1) (OI), which corresponds to $O^{2-}$ ions in the $Ga_2O_3$ lattice regions; (2) (OII), representing $O^{2-}$ ions in oxygen-deficient regions and commonly associated with oxygen-vacancy ($V_O$) defects in oxide materials; (3) (OIII) peaks, indicative of hydroxyl and carbonate species chemisorbed on the film surface, centered at 530.5, 532.0, and 533.5 eV, respectively [20]. The density of oxygen vacancies is indicated by the intensity ratio of $O_{II}/(O_{II} + O_I)$, and it is found to be 17.2%, 10.0%, 14.7%, 10.7%, and 10.1% for the AG, N-640, N-700, N-800, and N-900, respectively. The AG $Ga_2O_3$ film inherently exhibits a heightened concentration of oxygen vacancies, as evidenced by a substantial $O_{II}/(O_{II} + O_I)$ ratio of 17.2%. After 640 °C post-annealing, the $O_{II}/(O_{II} + O_I)$ ratio is significantly decreased to 10%, indicating the elimination of oxygen-vacancy defects in the film. Correspondingly, the Ga2p shifts in the high-binding-energy direction, as shown in Figure 3f. Therefore, the reduction in the $O_{II}/(O_{II} + O_I)$ ratio within the N-640 can be attributed to the increased formation of Ga-O bonds during annealing at 640 °C [27]. Conversely, with the 700 °C annealing, the $O_{II}/(O_{II} + O_I)$ ratio shows a notable increase, contrasting the behavior observed in the N-640 film. It is believed that the film crystalline phase begins to develop from ε phase to a mixture of ε and β phase at 700 °C annealing; meanwhile, the grain boundaries are formed inside the film. The sharp increase in $V_O$ concentration in the N-700 condition is primarily governed by the predominant escape of oxygen through grain boundaries or cavities [28]. Furthermore, when the annealing temperature is increased as high as 800 °C or 900 °C, the intensity ratio of $O_{II}/(O_I + O_{II})$ decreases to nearly 10%, while the Ga2p spectra shift in the low-binding-energy direction. Similar phenomena have been reported in the literature [29]. It was explained that N doping in β-$Ga_2O_3$ substitutes to Vo, during the high-temperature annealing process in a nitrogen atmosphere, while the Ga2p shift corresponds to the formation of Ga-N bonds. The impact of high-temperature annealing on point defects needs still further research.

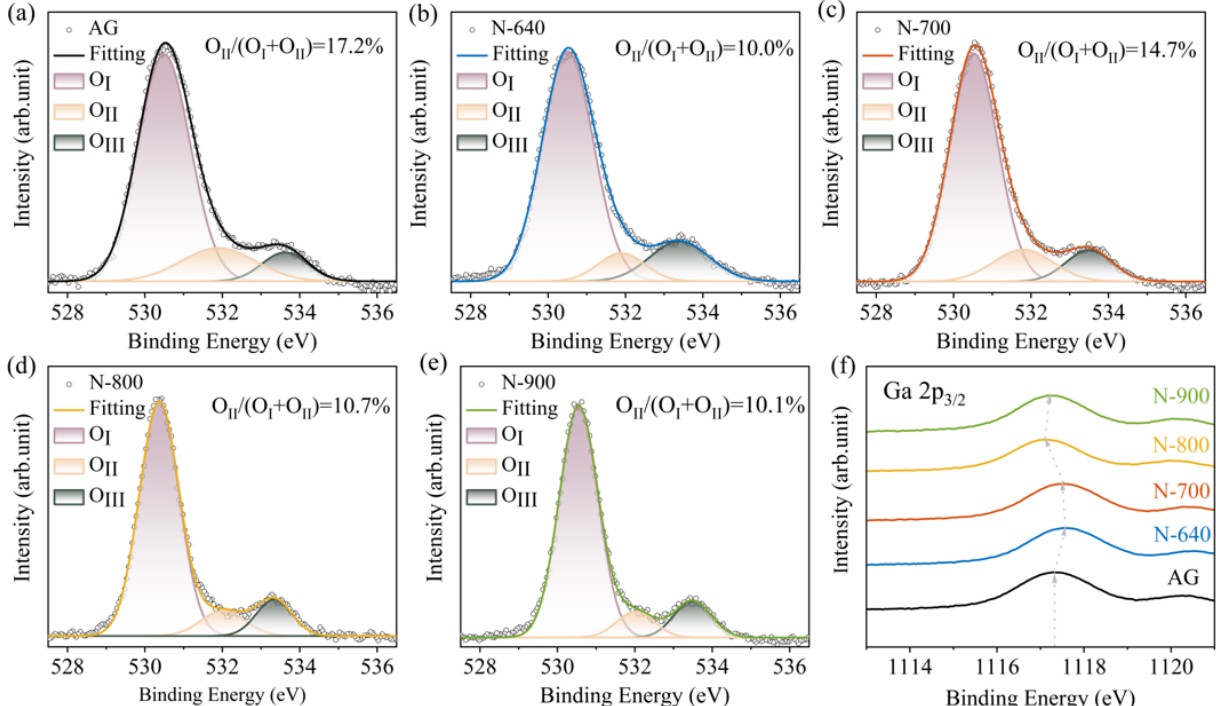

**Figure 3.** (**a–e**) XPS O 1s core level spectra of $Ga_2O_3$ films with various thermal annealing temperatures. (**f**) Ga $2p_{3/2}$ core-level spectra of AG, N-640, N-700, N-800, and N-900.

MSM SBPDs were fabricated on both the as-grown and annealed $Ga_2O_3$ thin films. In Figure 4a–e, the semilogarithmic I-V characteristics of all $Ga_2O_3$ SBPDs in the dark and under 254 nm illumination (power density ~62 $\mu W/cm^2$) at room temperature are presented. The primary distinctions in dark current ($I_{dark}$) and photocurrents ($I_{254nm}$) at 10 V were extracted, as illustrated in Figure 4f. The as-grown (AG) SBPD exhibits elevated leakage current attributed to the presence of oxygen-vacancy ($V_O$) defects within the $\varepsilon$-$Ga_2O_3$ film. According to some reports, in the MSM-type SPBDs, traps such as oxygen-vacancy defects under the conduction band promote trap-assisted tunneling (TAT) during the electron transport so that the device shows a large dark current [30,31]. However, the $I_{dark}$ at 10 V maintains an exceptionally low level, measuring below $6.9 \times 10^{-14}$ A after undergoing thermal annealing at 640 °C; this value is nearly three orders of magnitude lower than the dark current observed in the AG device. Considering that the crystal quality is almost unchanged after post-annealing at 640 °C, such a low $I_{dark}$ should be ascribed to the plunge in the concentration of point defects such as Vo. Then, the $I_{dark}$ rises with the increase in annealing temperature, even larger than that of the AG SBPD. It can be attributed to the more leakage channels caused by the worse crystalline quality. Of course, for the case with 700 °C thermal annealing, the oxygen-related defects also lead to the large leakage current. In contrast to the behavior of $I_{dark}$, the $I_{photo}$ remains consistently high across all devices. The combination of high $I_{photo}$ and low $I_{dark}$ results in an increased photo-to-dark current ratio (PDCR = ($I_{photo} - I_{dark}$)/$I_{dark}$). Notably, the N-640 $Ga_2O_3$ SBPD exhibits an exceptionally high PDCR of $3 \times 10^5$, surpassing all other devices.

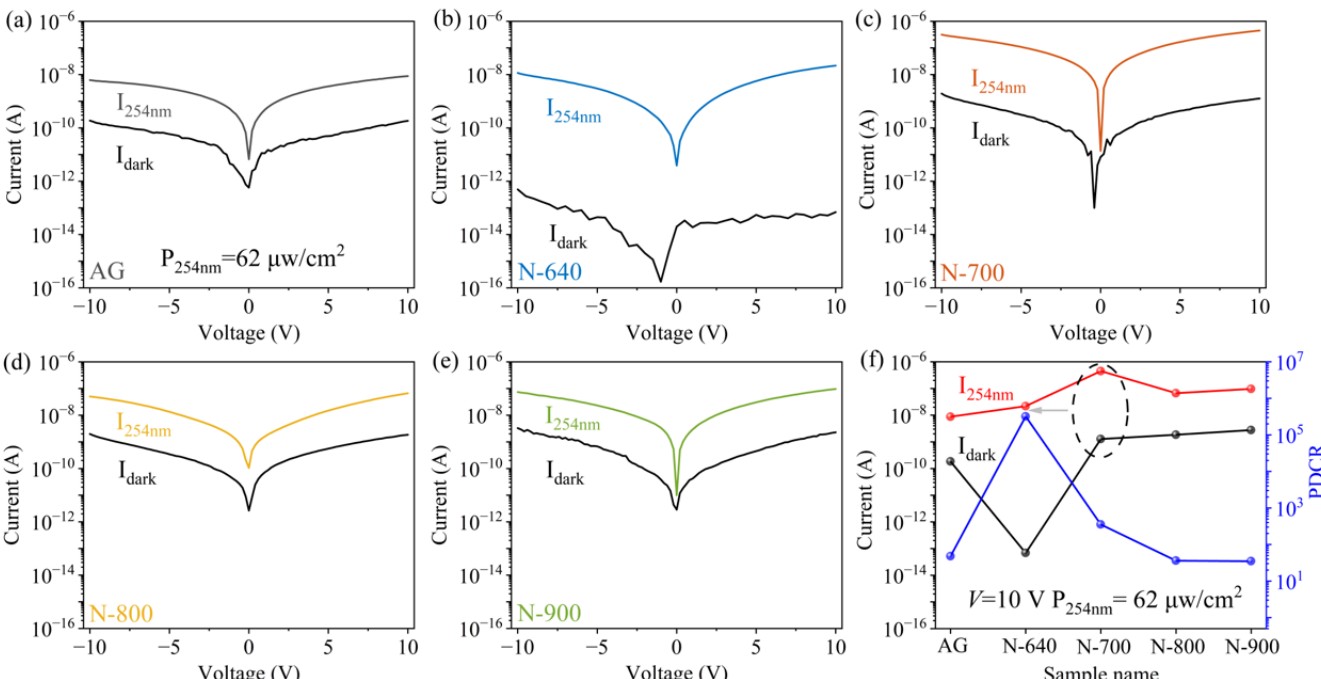

**Figure 4.** (**a–e**) Logarithmic scale I-V curves of the $Ga_2O_3$ SBPDs with and without 254 nm light (a power density of $P_{254\,nm} = 62\ \mu W\ cm^{-2}$). (**f**) The dependence of $I_{dark}$ & $I_{254\,nm}$ & PDCR on the various annealing temperatures (@ 10 V).

Figure 5a is the normalized spectral response of all the devices measured at a wavelength range of 220 to 400 nm and a bias of 10 V. Thermal annealing significantly impacts the spectral responsivity. Assessing the spectral response characteristics illustrated in Figure 5a, it is evident that all devices demonstrate substantial sensitivities in the solar-blind region. Upon undergoing a phase transition, the response peak of $Ga_2O_3$ SBPDs shifts from 230 to 256 nm. The crystal structure plays a pivotal role in influencing the response peak, presumably determining the energy band structure and the distribution of the density of states. The UV-to-visible rejection ratio ($R_{peak}/R_{400nm}$), representing the ratio of peak responsivity

to responsivity at 400 nm, serves as a metric for evaluating the spectral selectivity of the photodetector. As depicted in Figure 5b, the rejection ratio of the N-640 device can reach an impressive $2.4 \times 10^4$. This optimal wavelength selectivity and sensitivity suggest a significant reduction in subgap states throughout the entire $\varepsilon$-Ga$_2$O$_3$, achieved through thermal annealing at 640 °C. Furthermore, another critical figure of merit is $D^*$, which characterizes sensitivity, taking into account the noise floor from $I_{dark}$ in addition to the photo response, and is expressed as [32].

$$D* = \frac{R\sqrt{S_{eff}}}{\sqrt{2qI_{dark}}} \tag{1}$$

Here, $R$ represents the responsivity, $S_{eff}$ is the effective illuminated area in the detector, and $q$ denotes the electron charge ($1.6 \times 10^{-19}$ C). As illustrated in Figure 5b, the specific detectivity of the N-640 photodetector is determined to be $1.0 \times 10^{15}$ cm Hz$^{1/2}$ W$^{-1}$ (Jones), attributed to its high responsivity and low dark current. Figure 5c depicts the responsivity of all the devices under illumination of various 254 nm light intensities. It is shown that the responsivity exhibits a trend of initial increase followed by a decrease with increasing light intensity. Under low light intensity illumination, the photoconductivity effect constitutes the primary operational principle of the device. Trap centers capture and subsequently release photogenerated charge carriers, prolonging their lifetimes and thereby enhancing responsivity. Conversely, under high light intensity illumination, the self-heating effect at elevated intensities leads to increased carrier scattering, resulting in a decrease in responsivity [33,34].

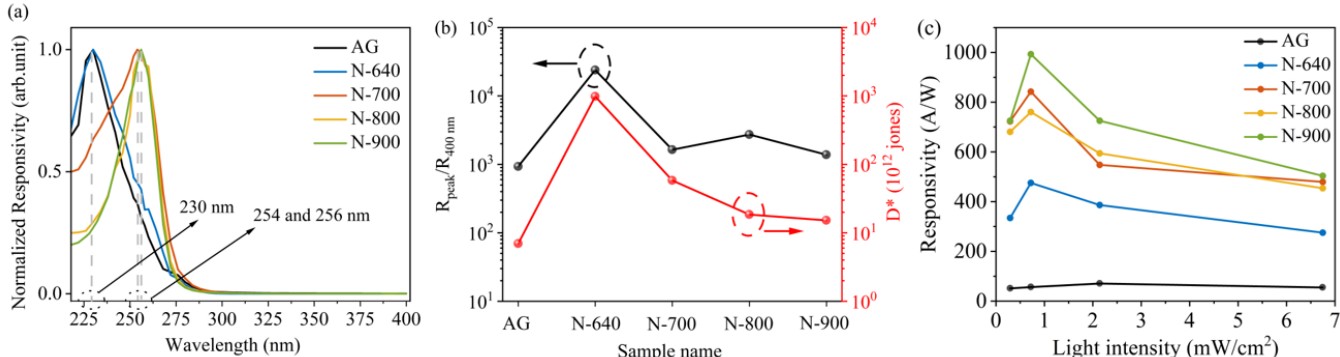

**Figure 5.** (**a**) The normalized spectral responsivity of all the devices (@ 10 +V) (**b**) The dependence of the UV-to-visible rejection ratio (R$_{peak}$/R$_{400nm}$) and specific detectivity ($D^*$) of all the devices. (@ 10 +V) (**c**) Light intensity-dependent responsivity of the AG, N-640, N-700, N-800 and N-900 devices (@ 10 +V).

Figure 6a exhibits the time-dependent photo-response of all the photodetectors, acquired by exposing them to 254 nm UV light at a power density (P$_\lambda$) of 550 μW/cm$^2$, while operating at a voltage of 10 V. All the devices exhibit a distinct photo-response at this light intensity. It is evident from Figure 6b that the decay time ($\tau_d$), defined as the duration during which the current decreases from 90% to 10% of its peak value, was significantly reduced by thermal annealing at 640 °C. Moreover, the decay time ($\tau_d$) typically consists of two processes characterized by fast and slow responses. Typically, the fast response components stem from immediate shifts in carrier concentration upon light activation or deactivation, while the slow response is associated with carrier trapping and release from defect bands like oxygen vacancies, gallium–oxygen-vacancy pairs, or grain boundaries within Ga$_2$O$_3$ films [35,36]. For a quantitative analysis of the switching speed of the PDs, the time-dependent response curves (see Figure 6a) were subjected to fitting using a bi-exponential relaxation equation [37], as outlined below:

$$I = I_0 + Ae^{-t/\tau_1} + Be^{-t/\tau_2} \tag{2}$$

Here, $I_0$ represents the steady-state photocurrent, t denotes time, and *A*, *B* are constants for fitting. $\tau_{d1}$ and $\tau_{d2}$ represent the relaxation time constants for the fast and slow components, respectively. Figure 6c shows the decay time of all the devices. They ($\tau_{d1}$ and $\tau_{d2}$) are 0.08 s/0.37 s, 0.03 s/0.11 s, 0.05 s/0.53 s, 0.05 s/0.48 s, and 0.07 s/0.58 s for AG, N-640, N-700, N-800, and N-900, respectively. The persistent photoconductivity (PPC) has been widely reported, which is mainly related to deep-level defects, which can come from dislocations, grain boundaries, as well as point defects such as Vo. Obviously, the decay speed can be effectively accelerated with 640 °C thermal annealing. The improved persistent photoconductivity (PPC) can be attributed to superior crystal quality and a decreased presence of oxygen vacancies. As depicted in Figure 6d, the N-640 photodetector exhibits a time-dependent photo-response to 254 nm illumination through on/off switching under an applied bias of 10 V. Even after multiple illumination cycles, the device consistently maintains a nearly identical response, emphasizing its high robustness and excellent reproducibility. Table 1 summarizes the parameters of the typical $Ga_2O_3$ MSM-type SBPDs with mainstream phases. In this work, the dark current of the N-640 photodetector is $\sim 6.9 \times 10^{-14}$ A. For $\varepsilon$-$Ga_2O_3$ SBPDs, this is the lowest dark current reported in the literature so far. Moreover, the PDCR, responsivity, and $\tau_d$ exhibit comparability to, or slightly surpass, results reported in other works.

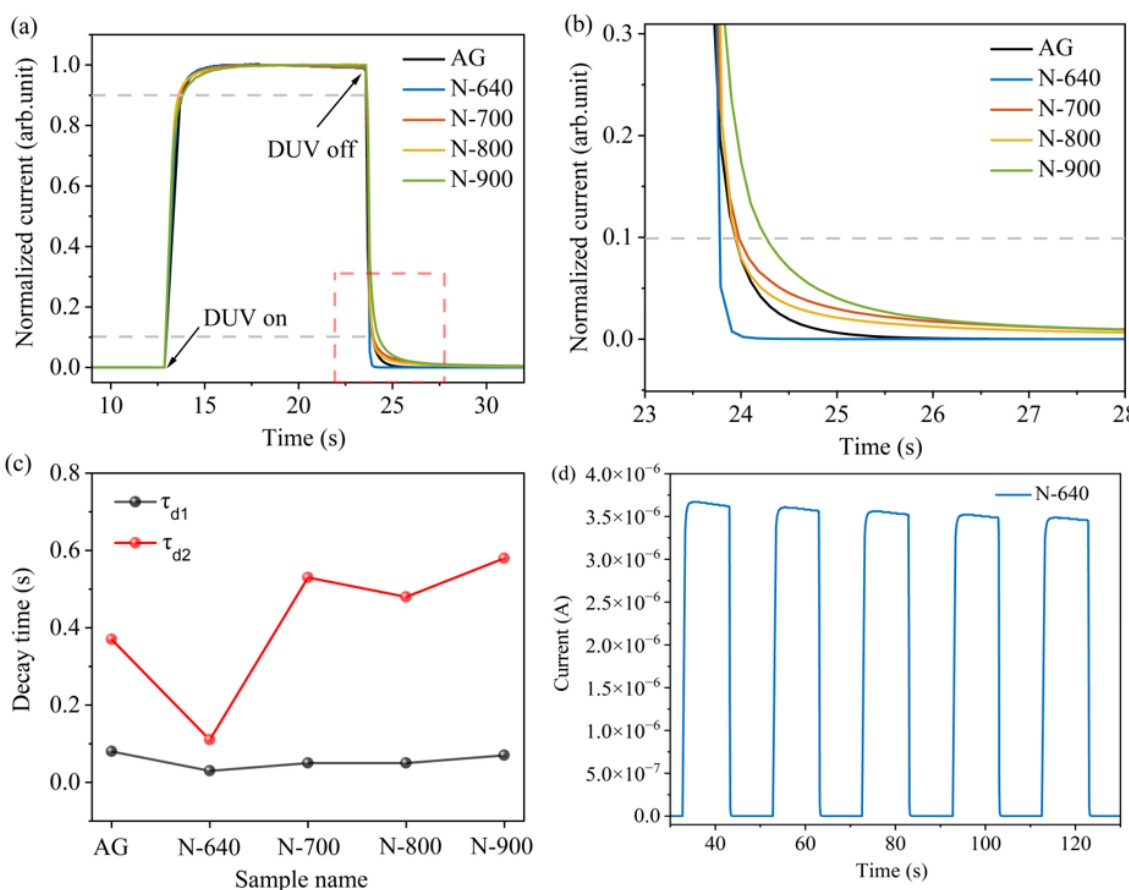

**Figure 6.** (**a**) Normalized I-t curves of the photodetectors. (**b**) Enlarged view of the slow response component of the decay edge of SBPDs. (**c**) $\tau_{d1}$ and $\tau_{d2}$ of all the devices. (**d**) Photoresponse of the N-640 by switching 254 nm light.

**Table 1.** Comparison of the device parameters of the reported $Ga_2O_3$ thin film MSM SBPDs.

| Materials | Bias (V) | PDCR | $I_{dark}$ (A) | D* (Jone) | Rejection Ratio | $\tau_d$ (s) | Reference |
|---|---|---|---|---|---|---|---|
| $\varepsilon$-$Ga_2O_3$ | 10 | $3.0 \times 10^5$ | $6.9 \times 10^{-14}$ | $1.0 \times 10^{15}$ | $2.4 \times 10^4$ ($R_{230}/R_{400}$) | 0.03/0.11 | This work |
| $\varepsilon$-$Ga_2O_3$ | 5 | $8.7 \times 10^6$ | $1.5 \times 10^{-12}$ | $2.5 \times 10^{15}$ | $1.2 \times 10^6$ ($R_{250}/R_{400}$) | 0.13 | [38] |
| $\varepsilon$-$Ga_2O_3$ | 5 | $3.5 \times 10^3$ | $2.1 \times 10^{-11}$ | $2.5 \times 10^{11}$ | - | - | [39] |
| $\varepsilon$-$Ga_2O_3$ | 6 | $1.7 \times 10^5$ | $2.4 \times 10^{-11}$ | $1.2 \times 10^{15}$ | $1.2 \times 10^5$ ($R_{250}/R_{400}$) | 0.03/0.08 | [40] |
| $\varepsilon$-$Ga_2O_3$ | 6 | $5.7 \times 10^4$ | $2.5 \times 10^{-11}$ | $4.2 \times 10^{14}$ | - | 0.10 | [41] |
| $\varepsilon$-$Ga_2O_3$ | 5 | $2.0 \times 10^3$ | $4.7 \times 10^{-7}$ | $1.2 \times 10^{13}$ | $6.0 \times 10^3$ ($R_{254}/R_{400}$) | 5.20 | [42] |
| $\varepsilon$-$Ga_2O_3$ | 20 | $9.5 \times 10^7$ | $1.0 \times 10^{-12}$ | $1.0 \times 10^{16}$ | $1.9 \times 10^4$ ($R_{240}/R_{400}$) | 0.12 | [43] |
| a-$Ga_2O_3$ | 10 | $<10^4$ | $3.4 \times 10^{-10}$ | $1.3 \times 10^{14}$ | $1.2 \times 10^5$ ($R_{250}/R_{350}$) | 0.02/0.35 | [20] |
| a-$Ga_2O_3$ | 5 | $1.0 \times 10^5$ | $3.0 \times 10^{-15}$ | $9.8 \times 10^{12}$ | $1.0 \times 10^3$ | $2.40 \times 10^{-4}$ | [31] |
| $\beta$-$Ga_2O_3$ | 20 | $1.5 \times 10^5$ | $7.0 \times 10^{-10}$ | $1.1 \times 10^{16}$ | - | 0.21/1.38 | [44] |
| $\beta$-$Ga_2O_3$ | 10 | $<10^7$ | $2.9 \times 10^{-11}$ | $9.0 \times 10^{15}$ | $8.6 \times 10^6$ ($R_{250}/R_{400}$) | 0.50 | [45] |
| $\beta$-$Ga_2O_3$ | 50 | $3.5 \times 10^1$ | $1.0 \times 10^{-9}$ | - | - | 12.13 | [46] |
| $\alpha$-$Ga_2O_3$ | 10 | - | $5.0 \times 10^{-13}$ | - | $3.6 \times 10^2$ ($R_{253}/R_{400}$) | $8.90 \times 10^{-5}$ | [47] |
| $\alpha$-$Ga_2O_3$ | 10 | $1.0 \times 10^2$ | $1.2 \times 10^{-9}$ | - | - | 3.49/5.12 | [48] |
| $\gamma$-$Ga_2O_3$ | - | $1.6 \times 10^4$ | $9.0 \times 10^{-10}$ | - | - | 0.06 | [49] |

## 4. Conclusions

In summary, we have showcased MSM SBPDs based on $\varepsilon$-$Ga_2O_3$ thin films, thoroughly exploring the impact of post-annealing on device performance. Notably, the solar-blind photodetector, subjected to a 640 °C thermal annealing process under a nitrogen atmosphere, demonstrates superior performance parameters: a low dark current of ~$6.9 \times 10^{-14}$ A at 10 V bias, a high PDCR of $3 \times 10^5$, and a substantial rejection ratio ($R_{peak}/R_{400}$) of $2.4 \times 10^4$ ($R_{peak}$ = 230 nm). The excellent performance can be ascribed to the diminished presence of oxygen-related vacancy defects following annealing, all the while preserving a high crystal quality. It is crucial to note that when the post-annealing is performed at high temperature, the $\varepsilon$-$Ga_2O_3$ transforms to $\beta$-$Ga_2O_3$ with worse crystal quality and the performance of SBPDs significantly decreases. Our research underscores the importance of employing an appropriately tailored thermal annealing process as an efficient strategy to achieve high-performance $\varepsilon$-$Ga_2O_3$-film-based SBPDs.

**Supplementary Materials:** The following supporting information can be downloaded at: https://www.mdpi.com/article/10.3390/coatings13121987/s1, Figure S1. XRD patterns of the $\varepsilon$-$Ga_2O_3$ seeding layer growth on c-plane sapphire.

**Author Contributions:** Z.F.: Methodology, Investigation, Validation, Formal analysis, Writing—original draft. Z.C.: Conceptualization, Methodology, Formal analysis, Writing—review & editing. W.C.: Investigation, Formal analysis. T.L.: Investigation. S.C.: Investigation. J.L.: Conceptualization, Funding acquisition. X.W.: Conceptualization, Methodology. X.L.: Conceptualization, Methodology. G.W.: Conceptualization, Methodology. Y.P.: Conceptualization, Methodology, Formal analysis, Writing—review & editing, Supervision, Funding acquisition. All authors have read and agreed to the published version of the manuscript.

**Funding:** This work was supported in part by the Natural Science Foundation of China (Grant No. 61804187 and No. 62074167), the Science and Technology Development Plan Project of Jilin Province, China (Grant No. YDZJ202303CGZH022), and the Open Fund of the State Key Laboratory of Optoelectronic Materials and Technologies (Sun Yat-sen University).

**Institutional Review Board Statement:** Not applicable.

**Informed Consent Statement:** Not applicable.

**Data Availability Statement:** The data presented in this study are available on request from the corresponding author.

**Acknowledgments:** We acknowledge Runze Zhan of the State Key Laboratory of Optoelectronic Materials and Technologies for XPS measurements.

**Conflicts of Interest:** The authors declare no conflict of interest.

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
