# Peer review of "High-Performance ε-Ga2O3 Solar-Blind Photodetectors Grown by MOCVD with Post-Thermal Annealing"

_coatings, doi:10.3390/coatings13121987_

Round 1

Reviewer 1 Report

Comments and Suggestions for Authors

There are several moments keeping me straight away to accept the current manuscript fpr publications:

1) In introductionpart, the authors claimed that there is no article devoted to influence of the post-annealing on the resulted e-Ga2O3 based solar-blind photodetector. However, it is not correct. Please check "Oxygen vacancies modulating the photodetector performances in e-Ga2O3 thin films; DOI https://doi.org/10.1039/D1TC00616A" article. Moreover, the basic principle and mechanism of the oxygen defects midiated photoresponse of the epsilon phase of Ga2O3 have been published in numerous researches. That is why, authors should provide more solid arguments to be considered for publications. 

2)  Concerning XRD results, I have found no data for a seeding layer of Ga2O3 that in my opinion has a crucial influence at the second step of the MOCVD grown e-Ga2O3. 

3) Authors claimed that AG and N-640 samples both are pure e-Ga2O3 thin films. However, the XRD results presented showd a distinct shoulders near 36 and 58 degrees which are the contribution of the beta-Ga2O3 phase. That is why, authors did not obtain pure monophase e-Ga2O3. Thus, speculating about defects such as oxygen vacancy and so on is incorrect before excluding the influence of other competing and more stable b-Ga2O3 phase.

4) Surface morphology analysis presented in Figure 2 is not informative at all. I reccomend to perform AFM measurements at least.

5) In dealing with defect mediated photoresponse characteristics of the solar-blind photodetector, a photoresponse vs. incident light intensity is mandatory.

6) The text contents too much techical errors and English grammar imperfections.

That is why, my overall reccomendation is to perform a major revision taking into account all the remarks above.  

Comments on the Quality of English Language

line 12: heteroepitaxial = heteroepitaxially

line 23: photo-response => photoresponse

line 34: because its => because of its

line 37: crystal anisotropy has no direct influence on heteroepitaxy. Consider lattice parameters and thermal expansion coefficient instead in comparisson between epsilon and beta phases.

lines 38-39: which makes e-Ga2O3 to be easily grown...

line 44: term "multifaceted" is unappropriate sinse it may be misunderstood in view of its crystalographic origin. Consider instead: ...on performance can appear in a variety of ways.

line 58: "anaerobic" term is related to metabolism, while oxygen-free is the correct way here

line 62: temperatures

line 68: double side polishd

line 70: TEGa should be identified before abbreviation

line 80: electrodes

line 84: (I-t)

line 87: as a light source

line 88: light modulation parameters should be provided

line 76: X-ray radiation source should be provided

line 92: crystal planes with negative Miller index should be rewritten in a correct way

line 153: semi-logarithmic

Reviewer 2 Report

Comments and Suggestions for Authors

In this manuscript, the authors have investigated the effects of post-annealing on ε-Ga2O3 metal-semiconductor-metal (MSM) type solar-blind photodetectors (SBPDs). The performance of the SBPD with the post-annealing of 640 °C is improved greatly compared to the ones fabricated on the other films. When the post-annealing is performed at high temperatures, the ε-Ga2O3 transforms to β-Ga2O3 with worse crystal quality, and the performance of SBPDs significantly decreases. I recommend this manuscript be published, but some minor points need to be addressed to improve the paper's quality:

1.     The Abstract needs to be attractive with emphasize your novelty.

2.     One or two sentences need to be added at the beginning of the introduction about photodetector concepts and work mechanisms, then SBPDs and MSM.

3.     The quality of Figs. 3 and 4 need to be improved. 

Reviewer 3 Report

Comments and Suggestions for Authors

Title: Well-relate to the presented work.

This work presents investigation on ε-Ga2O3 thin films grown on sapphire substrates. Post-annealing reduced oxygen-related defects, improving the crystal quality. Solar-blind photodetectors (SBPDs) fabricated from these films showed enhanced performance, including low dark current, high rejection ratio, and improved photo-to-dark current ratio. The study demonstrates the effectiveness of post-annealing in eliminating defects and enhancing SBPD performance.

Introduction: para 1, line 2, “missile warming”- correct to missile warning

Line 3: “fileds” to fields

Analysis does not include the determination of band gap of the material. It would be interesting if the value can be provided.

References. Most of the references quite old… more than 5 years. It’s good to refer to the latest (less than 5 years) to show relevancy.

Overall, a good written manuscript.

Comments on the Quality of English Language

English is ok, but can further be improved.

Reviewer 4 Report

Comments and Suggestions for Authors

In the manuscript the results of investigation of photoelectric properties of solar-blind photodetectors fabricated on ε-Ga2O3 thin films grown by metal-organic chemical vapor deposition are presented. The effect of the different annealing temperature under nitrogen atmosphere on the quality of the ε-Ga2O3 62 thin films and performance of solar-blind photodetectors was studied. It was found that annealing at 640 °C significantly improved performance of solar-blind photodetectors.  The time-dependent photo-response measurements show that decay time of current for the photodetectors annealed at640 °C is shortest 0.03 s/0.11 s.

It is unclear why photocurrent does not reach a stationary value within 10s. (See Figure 6 (d)). If, in fact, slow decay time is of the order of 0.11s, the photocurrent would reach a stationary value within 1.5s.

The manuscript must be amended.

Round 2

Reviewer 1 Report

Comments and Suggestions for Authors

Authors have adressed all my comments in the revised manuscript